# Data Fusion-Based Musculoskeletal Synergies in the Grasping Hand

**DOI:** 10.3390/s22197417

**Published:** 2022-09-29

**Authors:** Parthan Olikkal, Dingyi Pei, Tülay Adali, Nilanjan Banerjee, Ramana Vinjamuri

**Affiliations:** Department of Computer Science and Electrical Engineering, University of Maryland Baltimore County, Baltimore, MD 21250, USA

**Keywords:** kinematic synergies, muscle synergies, musculoskeletal synergies, principal component analysis, independent component analysis, hand grasp, activities of daily living

## Abstract

The hypothesis that the central nervous system (CNS) makes use of synergies or movement primitives in achieving simple to complex movements has inspired the investigation of different types of synergies. Kinematic and muscle synergies have been extensively studied in the literature, but only a few studies have compared and combined both types of synergies during the control and coordination of the human hand. In this paper, synergies were extracted first independently (called kinematic and muscle synergies) and then combined through data fusion (called musculoskeletal synergies) from 26 activities of daily living in 22 individuals using principal component analysis (PCA) and independent component analysis (ICA). By a weighted linear combination of musculoskeletal synergies, the recorded kinematics and the recorded muscle activities were reconstructed. The performances of musculoskeletal synergies in reconstructing the movements were compared to the synergies reported previously in the literature by us and others. The results indicate that the musculoskeletal synergies performed better than the synergies extracted without fusion. We attribute this improvement in performance to the musculoskeletal synergies that were generated on the basis of the cross-information between muscle and kinematic activities. Moreover, the synergies extracted using ICA performed better than the synergies extracted using PCA. These musculoskeletal synergies can possibly improve the capabilities of the current methodologies used to control high dimensional prosthetics and exoskeletons.

## 1. Introduction

Dexterity is one of the most amazing gifts that nature has provided us. Even the very basic activities that we perform in our daily life with minimal dexterity pose a remarkably complex challenge when it comes to their replication by robots. Many interdisciplinary research studies are being conducted to address this challenge. The challenge is to find out how the central nervous system (CNS) can select appropriate groups of muscles to make a specific hand movement or respond to a stimulus with ease. The human hand has more than 20 degrees of freedom, which makes this challenge even more complex. Astoundingly the CNS has no difficulty in handling such complexity in controlling the human hand.

To describe how the CNS can effortlessly achieve complex hand movements, different hypotheses have been proposed by researchers such as the elimination hypothesis, the optimization hypothesis and the modularity hypothesis. Out of these, the modularity hypothesis could provide some key insights and answers to many questions that were unaddressed. The modularity hypothesis was introduced by Bernstein in 1967 [1]. He theorizes that a single variable named synergy controls a group of functional units, and each unit of this group is formed by CNS. This hypothesis addressed the challenge of control and coordination of hand movements with vast degrees of freedom.

Motivated by the modularity hypothesis suggested by Bernstein, many researchers have attempted to solve the degrees of freedom problem through different concepts of synergies such as postural synergies [2,3], kinematic synergies [4], dynamical synergies [5] and muscle synergies [6]. The complex interaction of neuromuscular processes leads to musculoskeletal movements, and consequently an action is performed. At the musculoskeletal level, two complex tasks are achieved efficiently by this human biomechanical system. First, a group of muscles are selected by the CNS to perform the task at hand. Studies related to muscle activity patterns such as [7] found that muscle activity patterns can be reconstructed through a weighted linear combination of a limited number of muscle synergies. Second, a group of skeletal finger joints are actuated to enact the task. Findings from [2,3] suggest that through a weighted linear combination of a limited number of kinematic synergies, angular velocities of finger joints can be reconstructed.

From foundational studies in synergy-based movement generation, it can be summarized that the CNS sends signals through the peripheral nervous system (PNS) to recruit kinematic and muscle synergies required for successful accomplishment of the task at hand [8,9,10,11]. This implies that planning for complex multidimensional movements and their corresponding muscle activities might happen at a lower dimensional space of synergies. With the use of dimensionality reduction methods, these kinematic and muscle synergies can be extracted under time-invariant and time-variant models. As the planning and recruitment of the synergies is hypothesized to occur at a low dimensional space, by using these models, a movement can be expressed as a weighted linear combination of synergies.

In this paper, we aim to address the objective efficacy of musculoskeletal synergies in the reconstruction of kinematic and muscle activities. This might provide better insights into the use of synergies in the field of neurorehabilitation and control of prostheses, exoskeletons and robotics. After careful evaluation and consideration, we selected the KIN-MUS UJI dataset, one of the largest available datasets with simultaneous hand kinematics and forearm muscle activities that focuses on activities of daily living (ADL) and instrumented activities of daily living (IADL) [12].

Although kinematic and muscle synergies were studied extensively, only a few of them have investigated the similarities between the recruitments of kinematic and muscle synergies. Using the time variant and time invariant models, the movement kinematics and muscle activities can be reconstructed as weighted linear combinations of kinematic and muscle synergies, respectively. By fusing muscle and kinematic activities together and then performing dimensionality reduction techniques, “musculoskeletal” synergies were extracted. Can such fusion enable collaboration between kinematic and muscle synergies? This formulation allows for kinematic and muscle synergies to inform each other about their covariant characteristics and nonlinear biomechanics. We hypothesize that the musculoskeletal synergies thus obtained using a fused dataset might have the potential to better reconstruct movements when compared to synergies extracted separately. Prior to fusion, the muscle activities from flexor muscles were treated as positive and extensor muscles were treated as negative corresponding to kinematic activities.

## 2. Methods and Analysis

### 2.1. Experiment

A publicly available dataset was used for data analysis in this paper. This publicly available dataset KIN-MUS UJI [12] consists of twenty-two right-handed subjects of which 12 are males and 10 are females with a mean age of 35 ± 9 years. All the subjects had no prior upper limb movement disorders. Prior to the experiment, all participants were required to provide a written informed consent. All the experiments were conducted in alignment with the rules and regulations of the Ethics Committee of the Universitat Jaume I, Spain. In order to check the ability and quality of hand while performing activities of daily living, the Sollerman Hand Function Test (SHFT) was performed.

The following 26 activities of daily living (ADL) tasks were performed by all subjects: collecting a coin and putting it into a change purse; opening and closing a zip; removing the coin from the change purse and leaving it on the table; catching and moving two different sized wooden cubes; lifting and moving an iron from one marked point to another; taking a screwdriver and turning a screw clockwise 360° with it; taking a nut and turning it until completely inserted inside the bolt; taking a key; placing it in a lock and turning it counterclockwise 180°; turning a door handle 30°; tying a shoelace; unscrewing two lids and leaving them on the table; passing two buttons through their respective buttonhole using both hands; taking a bandage and putting it on his/her left arm up to the elbow; taking a knife with the right hand and a fork with the left hand and splitting a piece of clay; taking a spoon with the right hand and using it 5 times to eat soup; picking up a pen from the table and writing his/her name and putting the pen back on the table; folding a piece of paper with both hands and placing it into an envelope and leaving it on the table; taking a clip and putting it on the flap of the envelope; writing with the keypad; picking up the phone, placing it to his/her ear and hanging up the phone; pouring 1 L of water from a carton into a jug; pouring water from the jug into a cup up to a marked point; pouring the water from the cup back into the jug; putting toothpaste on the toothbrush; using a spray over the table 5 times; and cleaning the table with a cloth for 5 s.

Hand movements were captured by the CyberGlove (CyberGloveSystems, San Jose, CA, USA) at a sampling rate of 100 Hz. Ten of the sensors that correspond to the metacarpophalangeal (MCP) and interphalangeal (IP) joints of the thumb and the MCP and proximal interphalangeal (PIP) joints of the other four fingers were used. Muscle activities were recorded by an 8-channel surface electromyography (SX230 sensors, Biometrics Ltd., Newport, UK) device at a sampling rate of 1000 Hz. The electrodes were placed in the seven most representative areas of the forearm to capture major muscle activities. These were: (i) Flexor carpi ulnaris (FCU); (ii) Flexor carpi radialis (FCR) and palmaris longus (PL); (iii) Flexor digitorum superficialis (FDS), Flexor digitorum profundus (FDP) and Flexor pollicis longus (FPL); (iv) Abductor pollicis longus (APL), Extensor pollicis longus (EPL) and brevis (EPB); (v) Extensor digitorum communis (EDC); (vi) Extensor carpi ulnaris (ECU); (vii) Brachioradialis (BR), Pronator teres (PT), Extensor carpi radialis brevis (ECRB) and longus (ECRL).

### 2.2. Preprocessing

The raw sensor data recorded from the CyberGlove were converted to joint angles. The conversion procedures performed were based on the non-linear calibration protocols discussed in [13]. Finally, the data were filtered with a second-order low-pass Butterworth filter with a threshold frequency of 5 Hz. This was further filtered with a Savitzky–Golay smoothing filter. For normalizing each joint for each task, for each subject, the sEMG data collected were normalized by the maximum sEMG value recorded from the posture matrix (mentioned in Section 2.3.(i)) from all the tasks. This was done in order to allow comparison of sEMG activity from the same area of different subjects or to compare sEMG activity between different areas. The sEMG records were ultimately filtered with a fourth-order bandpass filter between 25–500 Hz, rectified, filtered by a fourth-order low-pass filter at 8 Hz and smoothened by Gaussian smoothing. Both joint angles and sEMG datasets were then synchronized with the start and stop instants of each movement.

The data consisting of 26 ADL tasks were split into two sets: a training set with 16 tasks that were used for extraction of synergies and a testing set with 10 tasks that were used for testing the reconstruction with the extracted synergies. The two sets were split such that the tasks were distributed qualitatively to reflect the diversity of the tasks across training and testing.

### 2.3. Derivation of Synergies

In this paper, synergies were derived from hand kinematics and muscle activities. Then, these found synergies were used to reconstruct the testing data comprising of new hand kinematics and muscle activities, thus testing the generalizability of kinematic and muscle synergies. Mathematically, the movement kinematics or muscle activities can be expressed as a linear combination of principal components or synergies as illustrated below. Here, M is a muscle activity or movement kinematics with n sensors and a total of m samples. A represents the coefficients or the weight vector; k is the number of components or dimensions to be extracted from training data, and S is the synergy matrix with reduced dimensions.
(1)$$M_{n\times m}=A_{n\times k}\;\cdot \;S_{k\times m}$$

The above expression can be modelled either using a time-variant synergy model or a time-invariant synergy model. A time-varying pattern is generated by combining the synergies with time-varying scaling coefficients in the case of time variant models. A time-varying synergy represents the coordinated activation of a group of muscles at a specific time for each muscle. Different patterns can be thus generated by scaling the coefficient and shifting different synergies in time. This can be expressed as
(2)$$M\left(t\right)=\overset{N}{\underset{i=1}{\sum }}A_{i}\;\cdot \;S_{i}\left(t-t_{i}\right)$$

A time-invariant model combines synergies with fixed coefficients to generate a constant pattern as expressed below; here, N is the number of synergies, and $M\left(t\right)$ is the activation of muscles at time t.
(3)$$M\left(t\right)=\overset{N}{\underset{i=1}{\sum }}A_{i}\;\cdot \;S_{i}\left(t\right)$$

Although for the objectives both time invariant and time variant models were used, it was noted that time variant model outperformed time invariant model because of its flexibility. Therefore, in this paper, we will be using only the results from time variant model.

For determining the optimal number of synergies, we used the threshold method on the variance accounted for (VAF) curve. As suggested in [14], the number of synergies can be computed using the equation shown below, where λ represents the eigen value or the magnitude of the corresponding synergy; m represents the optimal number of synergies, and n represents the total number of synergies.
(4)$$\frac{\lambda_{1}+\lambda_{2}+\ldots +\lambda_{m}}{\lambda_{1}+\lambda_{2}+\ldots +\lambda_{n}}\;\geq 0.9$$

When the fraction of this eigen values reaches close to the threshold basis of 0.9 (90%) as suggested in [15], the corresponding *m* (*m* ≤ *n*) value signifies the optimal number of synergies.
(i)Kinematic and Muscle Synergies

To obtain kinematic synergies, first a posture matrix was prepared as discussed in [8]. This posture matrix has 16 columns corresponding to the 16 ADL tasks grouped under the training dataset. Each column was formed by cascading angular velocities of 10 hand joints as discussed under Section 2.1 Then, principal component analysis (PCA) was performed on this matrix to obtain PCs that account for maximum variance.

It was observed that the first 6 PCs were able to account for a variance greater than 90% on average from the VAF curve as illustrated in Figure 1. These 6 PCs were termed as kinematic synergies.

A similar procedure was repeated on muscle data to obtain muscle synergies. Here, a muscle activity matrix was prepared with 16 columns corresponding to 16 ADL tasks grouped under training dataset. Each column was formed by cascading normalized muscle activities from 7 muscle areas discussed under Section 2.1. From the VAF curve for muscle synergies, it was noted that the first 4 PCs were able to account for a mean variance greater than 90%. To enable the comparison between the usage of kinematic and muscle synergies in reconstruction, it was required to consider the same number of PCs. Thus, 6 PCs lead to 6 muscle synergies.

As an example, Figure 2 illustrates the maximum absolute values (MAV) of the first six kinematic and muscle synergies of subject 1. It can be observed from the figure that for the first three kinematic synergies, R-PIP is more dominant followed by M-PIP, P-PIP and I-PIP. For all kinematic synergies, it can be noted that M-MCP is more dominant than other MCP finger joints, whereas for the first three muscle synergies, ECU muscle is more dominant followed by EDU muscle, APL, EPL and EPB muscle group and FCU muscle compared to the other groups.
(ii)Musculoskeletal Synergies

With the normalized kinematic and muscle data available, feature standardization was performed on them as expressed below:(5)$$x^{\prime }=\frac{x-\bar{x}}{\sigma }$$

Here, $\bar{x}$ is the mean calculated from the normalized muscle or kinematic dataset; σ is the standard deviation, resulting in $x^{\prime }$ which is the new muscle or kinematic data.

Each sensor value of the normalized kinematic activities is passed through the above expression, resulting in kinematic activities with zero mean and unit variance. As mentioned in II.C(i), the angular velocity matrix is formed with 16 columns corresponding to the 16 ADL tasks grouped under the training data. Applying PCA on this matrix yields PCs or kinematic synergies. First 6 PCs accounted for 84% of total variance on average.

Normalized muscle activities are passed through the above equation, yielding muscle activities with zero mean and unit variance. The polarity for the extensor muscles was inverted corresponding to the negative polarity of kinematics for extension. Similar to II.C(i), muscle activity matrix was created with 16 columns corresponding to 16 ADL training tasks. PCs were obtained by applying PCA on the muscle activity matrix. These PCs are muscle synergies. First 3 muscle synergies made 92% of variance. From the VAF curve as illustrated in Figure 3, it can be noted that 6 muscle synergies contribute to 97% of the total variance.

For data fusion, normalized muscle activities were concatenated with the normalized kinematic activities. Prior to fusion, the polarity for the extensor muscles was inverted as previously explained. The resultant matrix has 16 columns corresponding to 16 tasks. For each task, 7 muscle activities and 10 joint angular velocities were concatenated. Each datum in this matrix was then passed through the above expression, yielding a fused dataset with zero mean and unit variance. Resulting PCs obtained after applying PCA were termed as musculoskeletal synergies. It was noted that the first 6 musculoskeletal synergies formed 88% of total variance. The musculoskeletal synergies thus obtained were then split into musculoskeletal kinematic synergies and musculoskeletal muscle synergies.

Illustrated in Figure 4 are the MAVs of joint angular velocities of 10 joints for first six musculoskeletal kinematic (left) and MAVs of musculoskeletal muscle activities of seven areas for first six muscle (right) synergies extracted from the training data of subject 1. It can be observed that for most of the musculoskeletal muscle synergies, the ECU muscle group is more dominant followed by the EDU muscle group and APL, EPL and EPB muscle group. Though the dominant muscle groups are the same when compared to muscle synergies in Figure 2, the influence of muscle groups other than ECU is greater here. Similarly, for most of the musculoskeletal kinematic synergies, it can be calculated that I-PIP, M-PIP, R-PIP, P-PIP and M-MCP are more dominant than other joints. Although, the same joints are dominant for kinematic synergies as well, in Figure 2, we can notice an increased recruitment of these joints for all musculoskeletal kinematic synergies. This might be because when the two modalities were fused, muscle data might be able to influence the kinematic data.

For the purpose of comparison of the reconstruction results, while using kinematic, muscle and musculoskeletal synergies, it was necessary to either use the same number of components or use same variance that is over a given threshold variance (≥90% here). Using the same number of components implies comparison using the same number of synergies which is ideal for this study. Therefore, throughout all objectives, it can be noted that only the first 6 PCs were used.

### 2.4. Reconstruction of Movement Kinematics and Muscle Activities Using l_1_-Norm Minimization

The synergies thus obtained can be used as templates for decomposing hand movements as mentioned in [4]. Investigations from [4] hypothesize that the CNS’s strategy is to selectively make use of a small number of synergies to generate movement. A matrix was created which contains the row vectors of the movement synergies and all their possible shifts.
(6)$$m_{row}=cB$$

Here, the matrix B can be generally considered as the bank which contains all the different shifted version of either the kinematic or muscle synergies. Thus, for any given movement profile $m_{row}$ and an available bank of template synergies B, there exist many possible coefficients c which satisfy the above equation. The notion here is that the CNS uses only a small set of synergies and a small number of coefficients to achieve movements. Hence, this representation can be considered as an $l_{1}$-minimization problem and as presented in [4] can be formulated as an optimization problem for finding the sparsest possible coefficients for movement generation.
(7)$$Minimize\;{\left\|c\right\|}_{1}+\frac{1}{\lambda }+\|c\;B-m_{row}{\|}_{2}^{2}$$
where ${\|}_{.}{\|}_{2}$ represents the $l_{2}$ norm or Euclidean norm of a vector, and λ is the regularization parameter. In both the objectives, we have set λ = 0.01 $\lambda_{max}$, where $\lambda_{\max}$ gives the $l_{\infty }$ norm of $2m_{row}B^{\prime }$.

Thus, solving this optimization problem results in a set of possible weights or coefficients (c) of synergies used in reconstructing the movements grouped under the testing data.
(i)Kinematic and Muscle Synergies

The 6 kinematic and 6 muscle synergies obtained (as shown in Figure 2) were then shifted in time to generate different versions of their synergies. Each kinematic synergy was of 3916 samples (3.916 s), and these synergies were shifted by 25 samples for each version. A total of 40 shifted versions were created, thus making 1000 sample (1.000 s) shifts for one synergy. A kinematic synergy bank matrix was created with these 40 shifted versions of the kinematic synergies. Similar steps were performed for creating a muscle synergy bank. Each muscle synergy was of 3814 samples (3.184 s), and these synergies were shifted by 25 samples, generating 40 different shifted versions of muscle synergies for the bank. Thus, a total of 1000 sample shifts were made for one muscle synergy accounting for all the possible versions.

With the test data consisting of the movement kinematics and muscle activities and banks of kinematic synergies and muscle synergies, the corresponding coefficients for generating the movements were computed using *l*_1_-minimization. The results produced 240 feasible coefficients for 6 kinematic synergies and 6 muscle synergies separately. Thus, 40 coefficients were found for each of the kinematic and muscle synergies. These coefficients were then combined with the corresponding synergies to reconstruct the movements grouped under test data.
(ii)Musculoskeletal Synergies

Banks containing 40 shifted versions of musculoskeletal-muscle synergies and musculoskeletal-kinematic synergies were developed as mentioned in Section 2.4.(i). With a bank of all possible versions of a synergy and a given test data, there exist several different coefficients. As a result, different coefficients can combine with the synergy bank to reconstruct a movement as expressed below, where X is the reconstructed test data; $c_{i}$ is the coefficient of the $i^{th}$ version of the synergy, and $B_{i}$ is the $i^{th}$ version of the synergy in the synergy bank:(8)$$X=c_{i}\cdot \;B_{i}$$

There are 10 test tasks under the testing dataset, and the bank has 40 shifted versions for each synergy. The reconstruction error between the recorded movements $\left(M_{i}\right)$ and the reconstructed pattern $\left(X\right)$ was determined as
(9)$$err=\frac{\sum_{i}{\left(M_{i}-X\right)}^{2}}{\sum_{i}M_{i}^{2}}$$

This evaluation criterion helps to establish the performance of each synergy while using the corresponding coefficients to generate the test tasks.

## 3. Results

### 3.1. Extraction and Reconstruction Using Kinematic and Muscle Synergies

From the movement kinematics and the muscle activities recorded from 16 ADL tasks, six muscle synergies and six kinematic synergies were extracted using PCA. On average, for all the subjects, the first kinematic synergy accounted for more than 55% of the total variance as shown in Figure 1. As mentioned in [16], in this paper as well, it was noted that, while the first synergies contributed to more than 50% of the total variance, the remaining variances were allocated across several synergies. This indicates that a relatively small set of synergies could sufficiently represent high-dimensional kinematics. The same number of muscle synergies were considered for the purposes of comparison. As such, from the VAF curve in Figure 1, it can be noticed that the first muscle synergy accounted for an average of 65% variance. The extracted six kinematic and six muscle synergies were then used in reconstructing muscle activities and kinematics of 10 ADL tasks grouped for testing. Example reconstructions observed for task 7 (Taking a clip and putting it on the flap of the envelope) were shown in Figure 5.

From the VAF curve in Figure 3, it can be observed that the first muscle synergy, on average, account for over 75% of the total variance. As for musculoskeletal synergies, the first synergy accounted for more than 55% of the total variance whereas for the kinematic synergies, it was noted at 45%. This variation could be attributed to the fusion of kinematics and muscle activities. A total of 10 DoF joint kinematics had the highest variance; then came the fused kinematic and muscle activities, and lastly, 7 DoF muscle activities had the least variance. The fused data set did not have the highest variance indicating that there was cross information between kinematic and muscle activities. The musculoskeletal synergies were then split into musculoskeletal-kinematic synergies and musculoskeletal-muscle synergies. Although using 6 synergies did not contribute to 90% of the total variance, as mentioned previously, we are using the same number of components for fair comparison.

For all subjects, ten ADL and IADL tasks were reconstructed using the muscle synergies, kinematic synergies, musculoskeletal-muscle synergies, and musculoskeletal-kinematic synergies. Reconstructions achieved by musculoskeletal synergies were compared with the recorded activities and reconstructions achieved by muscle synergies and kinematic synergies. Figure 4 shows the contributions of various muscles and joints in musculoskeletal synergies. In the Figure 6, a sample of reconstruction for Task 1 of subject 5 is illustrated.

Figure 7 illustrates the reconstruction errors obtained while reconstructing 10 ADL tasks using synergies obtained with fusion and without fusion for all subjects. Overall, fusion improved the performance of musculoskeletal-kinematic synergies. In other words, musculoskeletal kinematic synergies learnt from the covariance of muscle activities but not vice-versa. The musculoskeletal muscle synergies did not perform well when compared to muscle synergies.

### 3.2. Independent Component Analysis (ICA)

We compared the performance of PCA vs. ICA in extracting musculoskeletal synergies. The musculoskeletal activity matrix detailed in Section 2.3.(ii) was processed through ICA instead of PCA. With the aim to find those independent components that might be used more than others in the 16 tasks, reconstruction ICA [17] (rICA in MATLAB^®^) was performed on the musculoskeletal matrix. Since six components are made use of for each hypothesis, the first 6 independent components (ICs) were obtained from rICA. These six ICs were then shifted in time to create the bank of synergies following the steps mentioned in Section 2.4. With the bank and test tasks available, through $l_{1}$-minimization, appropriate coefficients that best reconstruct the test tasks were obtained. Figure 8 represents the reconstruction result of subject 5, test task 1 (opening and closing a zip) using ICs. For comparison, reconstruction using musculoskeletal synergies of the same task were used.

By using the first six ICs, 10 ADL and IADL test tasks were reconstructed. This was performed for all the subjects. To evaluate the performance of the reconstruction results, as mentioned in Section 2.4.(ii), a similar equation was used to find the difference between the reconstructed and recorded movements kinematics and muscle recordings. Mean and standard deviation of this reconstruction error was obtained while considering all the subjects. Figure 9 depicts the mean and standard deviation observed while reconstructing the test tasks for all the subjects. It can be noted that except for test task 5, reconstruction results through ICs showed improvement for musculoskeletal synergies.

## 4. Discussion

In this paper, 22 subjects and 26 ADL and IADL tasks were considered. These are considerably large compared to other similar studies including ours. The results obtained from this study allow the possibility to generalize the results for a larger population when performing similar tasks [12]. The tasks considered here adequately represent the activities of daily living. The ADL and IADL tasks considered in this study provide accurate representation of hand grasp for rehabilitation due to their relationship with daily activities and their movement completeness. Similar to [16,18], this paper also takes into account only the 10 most often used hand joints while extracting kinematic synergies. The analysis performed in this study considers the entire movement profile which includes reaching, manipulating and release phases that were usually not considered in previous studies despite these events being a critical part of coordination. Moreover, these ADL and IADL tasks were carefully chosen through the SHFT that estimates the hand function. The obtained results were in accordance with our previous studies [16,19] in terms of number of synergies and the reconstruction errors. The implementation of dimensionality reduction methods on a publicly available dataset of movement kinematics and muscle activities to extract kinematic and muscle synergies helps in comparing the implementation procedures and results from other research studies as well.

Several studies have demonstrated strong correlations between neural, muscle and kinematic synergies. In [6], it was shown that muscle synergies align with kinematic synergies. In [20] muscles synergies were used as a predictive framework for the EMG patterns of new hand postures. In [18], it was found that spinal motor neuronal activities exhibit a synergistic organization that could be reflected in the neural drive received by muscle synergies. In [10,11], synergy-based hand kinematics were decoded and reconstructed for electroencephalographic signals. Inspired by these studies, in this paper, we allowed for the fusion of two modalities: movement kinematics and muscle activities. This fusion encourages the collaboration of both activities thus promoting learning between each other. Overall, the results reflect that the musculoskeletal synergies obtained from such fusion perform better in reconstruction of movement kinematics. To our best knowledge, this is one of the few papers that applied data fusion in combining two modalities to extract fusion synergies.

In this paper, considering the bimodal properties of joint kinematics and unimodal properties of muscle activities, we performed fusion between these two modalities. By bimodal joint kinematics, we mean that the same joint can perform both flexion and extension; thus, the polarity of the joint velocities is positive for flexion and negative for extension. In contrast, the unimodal properties of muscle activities imply that the flexor muscles have positive polarity for their muscle action potentials and similarly extensor muscles have negative polarity for their muscle action potentials. Thus, when combining these two types of modalities with bimodal and unimodal properties, a critical step was added prior to fusion as explained in Section 2.3, that is changing the polarity of the extensor muscles to negative. Thus, when the fusion occurs, the extension in kinematics is strengthened by the extension reflected in muscle activities. Without this change in polarity, all RMS muscle activities remain positive for both flexor and extensor muscles, and in contrast, the kinematic activities remain positive only for flexion and negative for extension. Fusion, without taking this polarity into account, proved to be detrimental to the obtained synergies. Furthermore, as noted in the results, the fusion has improved the performance of musculoskeletal kinematic synergies but not the performance of musculoskeletal muscle synergies. This can be attributed to the fact that all the key joints involved in ADL and IADL movements were included, but only extrinsic muscles were included. Intrinsic muscles [21] that balance finger movements, perform abduction/adduction and are responsible for coupling MCP and PIP joints were neither recorded nor included in this study. Including these muscles could have improved the collaboration between kinematics and muscles to improve musculoskeletal muscle synergies. Preliminary results obtained using this approach were presented in [22].

Although non-negative matrix factorization (NMF) is widely used for the extraction of muscle synergies from EMG signals [23,24,25] for hand based movements, Shourijeh et al. [26] identified the limitations of NMF as unreliable and unrepeatable for EMG decomposition. They suggest that the concatenation, such as implemented in this paper, can improve both repeatability and reliability. In this paper, based on our previous studies [3,11,27,28] that used prominent dimensionality reduction techniques like PCA, SVD and ICA, we made use of PCA and ICA. In [29], the authors have developed a novel algorithm, mixed matrix factorization, for the extraction of synergies similar to musculoskeletal synergies derived in this paper. In the near future, we will test this algorithm to extract musculoskeletal synergies.

Independent component analysis (ICA), a widely used dimensionality reduction and blind source separation technique, can effectively make use of higher-order statistics (HOS) to identify an independent set of synergies. ICA and its recent powerful generalization to multiple datasets, independent vector analysis (IVA), can be formulated to enable multimodal fusion [30,31,32,33]. ICA is optimal (in the maximum likelihood sense) if the probability density functions (PDFs) of the estimated sources perfectly match the “true PDFs of the sources” which are often unknown. Hence, while estimating the de-mixing matrix, the PDFs of the sources are also estimated. An ICA algorithm that allows a flexible PDF matching capability can improve the efficiency of the estimated sources. In this paper, as a preliminary demonstration, we used rICA to derive independent set of synergies. rICA based musculoskeletal kinematic synergies outperformed the synergies obtained from PCA. Adali et al. have introduced new ICA and IVA techniques following their introduction in 2006 [30,34,35]. They have demonstrated the capability of these techniques for joint analysis of multimodal (across different modalities) and multiset (across different conditions within a modality) neural signals [31,36,37], which we will extend to use in multimodal fusion in the near future.

## 5. Conclusions

This paper presented simultaneous extraction of kinematic and muscle synergies through data fusion and dimensionality reduction techniques, PCA and ICA. By using data fusion, we have attempted to find interactions between kinematic and muscle synergies, evaluating if there are any covariant characteristics that can inform each other. We hypothesized and tested whether these covariant characteristics can improve synergy-based decoding of movement kinematics. The results show promise that fusion synergies can improve movement reconstruction. In the areas of neuroscience [38,39,40], rehabilitation [38,41,42], sports [43,44,45] and robotics [46,47], synergies have proven to be an efficient method for understanding motor control and motor learning of the artificial and paralyzed limbs from a reduced dimensional space. It remains to be seen if embedding these fusion synergies into the above areas can possibly enhance their performance.

## Figures and Tables

**Figure 1 sensors-22-07417-f001:**
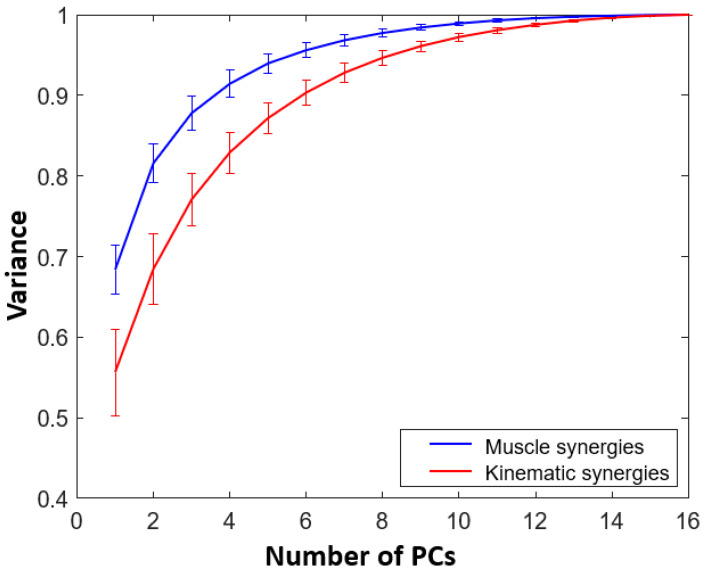
Variance accounted for (VAF) by muscle (in blue) and kinematic (in red) synergies (PCs). Mean and standard deviation are shown here. It can be noticed that six kinematic synergies and four muscle synergies cross the 90% threshold of the total variance.

**Figure 2 sensors-22-07417-f002:**
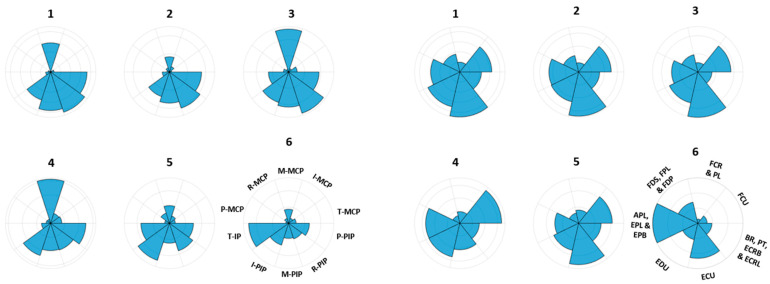
Maximum absolute values (MAV) of joint angular velocities of 10 joints for first six kinematic (**left**) and MAVs of muscle activities of seven areas for first six muscle (**right**) synergies extracted from the training data of subject 1 are shown here. The joints and muscles are defined in Section 2.1.

**Figure 3 sensors-22-07417-f003:**
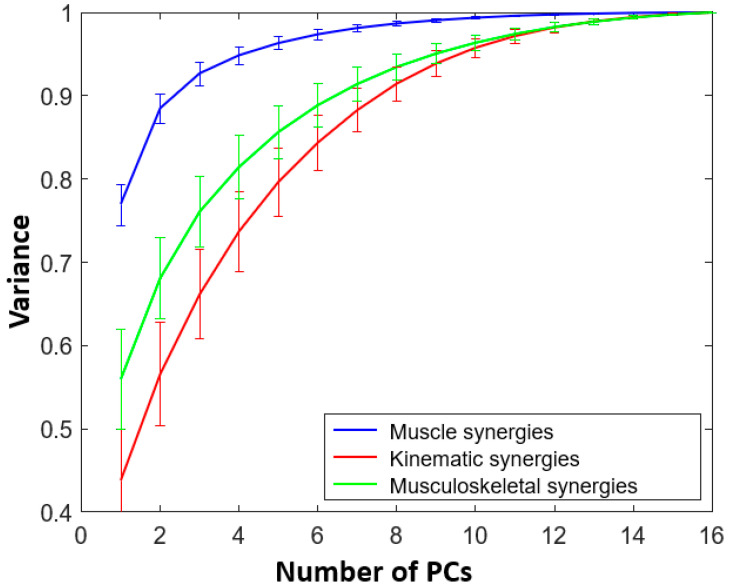
Mean of normalized muscle (in blue), kinematic (in red) and musculoskeletal (in green) variance for all subjects with standard deviation is exhibited here. It can be noted that eight kinematic synergies (or PCs), three muscle synergies and eight musculoskeletal synergies cross the 90% threshold of the total variance.

**Figure 4 sensors-22-07417-f004:**
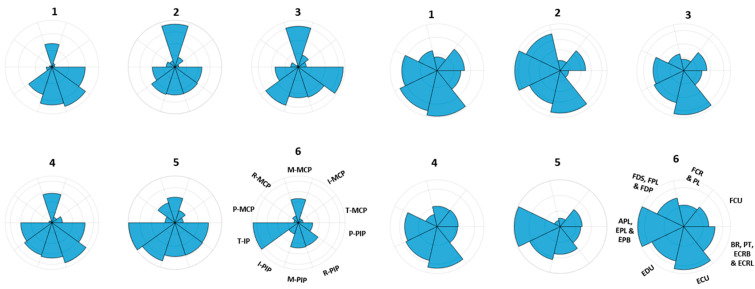
Maximum absolute values (MAV) of joint angular velocities of 10 joints for first six musculoskeletal kinematic (**left**) and MAVs of musculoskeletal muscle activities of seven areas for first six muscle (**right**) synergies, extracted from the training data of subject 1, are shown here. The joints and muscles are defined in Section 2.1.

**Figure 5 sensors-22-07417-f005:**
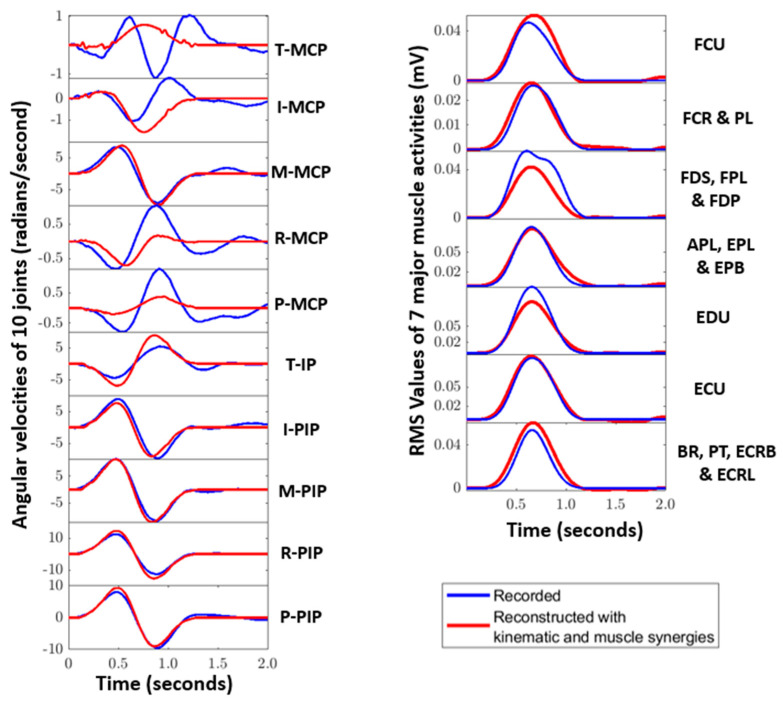
Reconstruction (red) of kinematic (**left**) and muscle (**right**) activities recorded (blue) for subject 1 for Task 7 (taking a clip and putting it on the flap of an envelope) is shown here. The subplots of the kinematic activities represent the 10 angular velocities of the finger joints, and the 7 subplots of the muscle activities represent the RMS values of the muscle groups detailed in Section 2.1.

**Figure 6 sensors-22-07417-f006:**
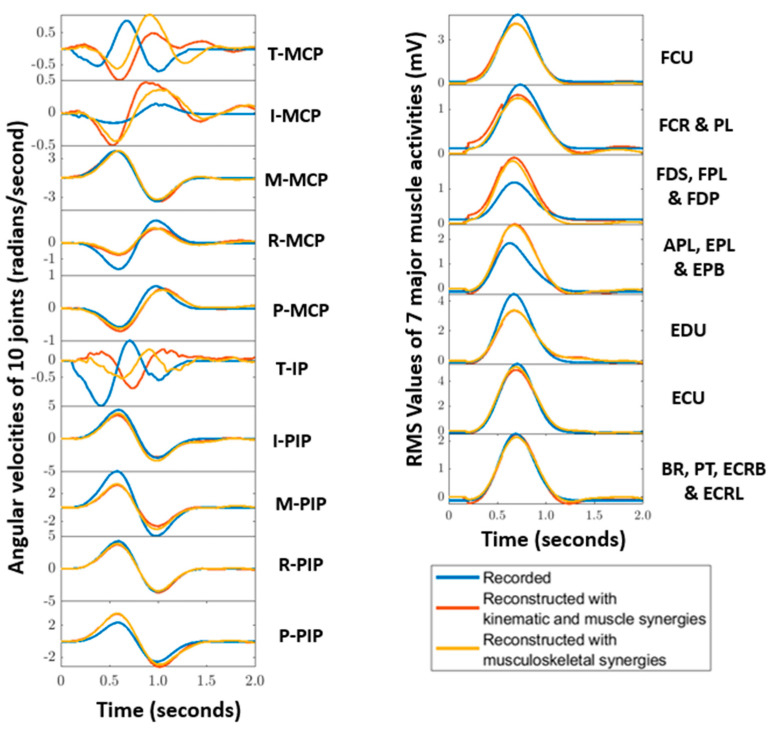
Reconstructions (red—with kinematic and muscle synergies, green—with musculoskeletal synergies) of movement kinematics (**left**) and muscle (**right**) activities recorded of subject 5 for Task 1 (opening and closing a zip) are shown here. The subplots of the kinematic activities represent the 10 angular velocities of the finger joints, and the 7 subplots of the muscle activities represent the RMS values of the muscle groups detailed in Section 2.1.

**Figure 7 sensors-22-07417-f007:**
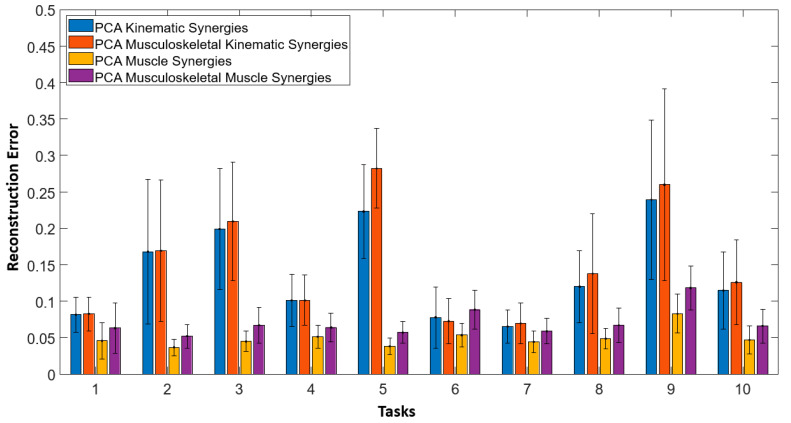
Reconstruction errors obtained while reconstructing 10 ADL tasks using synergies obtained with fusion (musculoskeletal-muscle and musculoskeletal-kinematic) and without fusion (muscle and kinematic) for all subjects are shown here. Overall fusion improved the performance of musculoskeletal-kinematic synergies. Bars indicate mean and errors bars indicate the standard deviation across all subjects.

**Figure 8 sensors-22-07417-f008:**
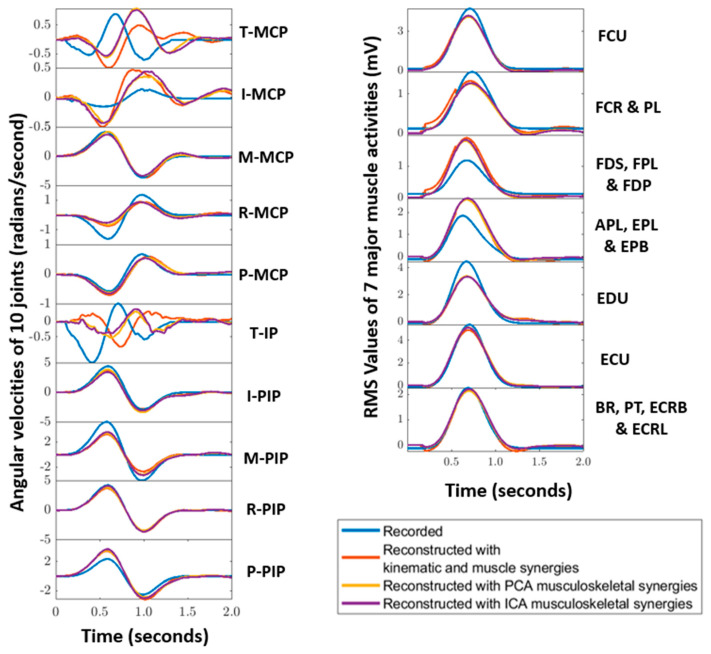
Reconstructions (red—with kinematic and muscle synergies, yellow—with PCA musculoskeletal synergies, and magenta—with ICA musculoskeletal synergies) of movement kinematics (**left**) and muscle (**right**) activities recorded (blue) of subject 5 for Task 1 (opening and closing a zip) are shown here. The subplots of the kinematic activities represent the 10 angular velocities of the finger joints, and the 7 subplots of the muscle activities represent the RMS values of the muscle groups detailed in Section 2.1.

**Figure 9 sensors-22-07417-f009:**
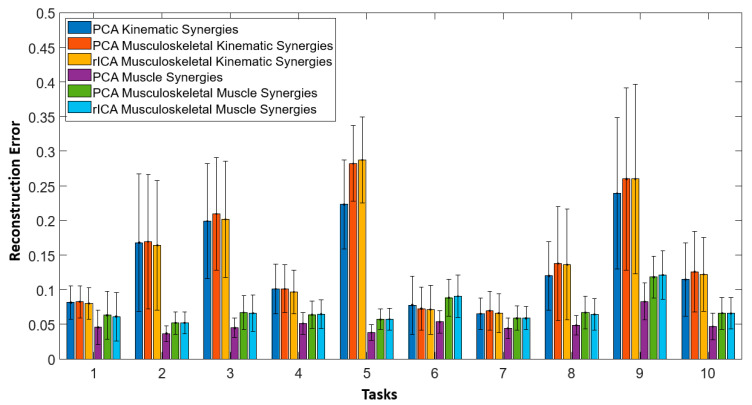
Mean reconstruction errors and standard deviations obtained while reconstructing the 10 ADL test task kinematics (using kinematic, musculoskeletal kinematic synergies) and test task muscle activities (using muscle, musculoskeletal muscle synergies) using PCA, fusion PCA and fusion rICA, respectively, are illustrated here.

## Data Availability

The data used in this paper are publicly available here [12]: N. J. Jarque-Bou, M. Vergara, J. L. Sancho-Bru, V. Gracia-Ibáñez and A. Roda-Sales, “A calibrated database of kinematics and EMG of the forearm and hand during activities of daily living,” *Scientific Data*, vol. 6, no. 1, pp. 1–11, Dec. 2019, https://doi.org/10.1038/s41597-019-0285-1.

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
