# Peer review of "Data Fusion-Based Musculoskeletal Synergies in the Grasping Hand"

_sensors, 2022, doi:10.3390/s22197417_

Round 1

Reviewer 1 Report

This study combines muscle and kinematic synergies. As claimed, this combination (called musculoskeletal synergies) performs better in the reconstruction of movement data. I have a major comment here about the clarity of this method. Unless there are enough constraints in the technique, decomposition approaches are not perfectly repeatable. For example, non-negative matrix factorization (NMF), has shown to be unreliable and unrepeatable for EMG decomposition. However, it has been shown that concatenation can improve both repeatability and reliability of such decomposition. Shourijeh, Mohammad S., Teresa E. Flaxman, and Daniel L. Benoit. "An approach for improving repeatability and reliability of non-negative matrix factorization for muscle synergy analysis." Journal of Electromyography and Kinesiology 26 (2016): 36-43. Although the methods differ and PCA is not as unreliable as NMF, I would like the authors to first clarify how reliable and repeatable PCA, ICA, and so on are on the date they used. Second, how do these two metrics change when muscle and kinematic data are combined? This would help to frame the work and its contribution more clearly. Here are minor comments: EMG normalization is critical in the decomposition of EMG synergies. How do the authors envision their approach being used in practice, when true maximum values are unavailable and EMGs cannot be truly normalized? Along the same lines, please clarify "for the particular area" in "The sEMG data collected was normalized by the maximum sEMG values recorded for that particular area for each subject." For all movements, EMG data were low-pass filtered at 8 Hz. I'm not sure why. I'm also curious how this will affect the study's findings. Please clarify. Lines 139-141: how were data divided into two sets (randomly)?

Reviewer 2 Report

In this paper, synergies were extracted first independently and then combined through data fusion using PCA and ICA. By a weighted linear combination of musculoskeletal synergies, the recorded kinematics and muscle activities were reconstructed. The results indicate that the musculoskeletal synergies performed better than the synergies extracted without fusion in. Also, the synergies extracted using ICA performed better than the synergies extracted using PCA. These musculoskeletal synergies can possibly improve the capabilities of the current methodologies used to control high dimensional prosthetics and exoskeletons, which is very interesting and worthy of discussion.

But there are still the following suggestions for authors:

1.     It is recommended to add some details description of data, such as the specific input and output of kinematic and muscle synergies.

2.     In the equation located in line 171, the variables  and  are not clear. Please add some description.

3.     Explain the purpose of Figure 2 and Figure 4 here.

4.     It is suggested to use a diagram to describe the relationship between several synergies and their input-output and fusion relationships.

5.    In line 158, how to obtain the  values at different times?

6.     Where can this musculoskeletal synergy be applied? How does the robotic hand replicate?

Round 2

Reviewer 2 Report

I think this paper is acceptable.

In this paper, synergies were extracted first independently and then combined through data fusion using PCA and ICA. By a weighted linear combination of musculoskeletal synergies, the recorded kinematics and muscle activities were reconstructed. The results indicate that the musculoskeletal synergies performed better than the synergies extracted without fusion in. Also, the synergies extracted using ICA performed better than the synergies extracted using PCA. These musculoskeletal synergies can possibly improve the capabilities of the current methodologies used to control high dimensional prosthetics and exoskeletons, which is very interesting and worthy of discussion.

Thank you very much for the authors’ clear explanation of my previous questions.

The explanation of some letters in the equation is also more detailed for readers to understand.

The comparison between Figure 2 and Figure 4 here can help readers better understand the influence from different muscle groups for musculoskeletal muscle synergies and the influence from different joints for musculoskeletal kinematic synergies.

Author Response

We have now revised the English language and style and revised any spelling mistakes in the manuscript.